# Imbalance between Actin Isoforms Contributes to Tumour Progression in Taxol-Resistant Triple-Negative Breast Cancer Cells

**DOI:** 10.3390/ijms25084530

**Published:** 2024-04-20

**Authors:** Vera Dugina, Maria Vasileva, Natalia Khromova, Svetlana Vinokurova, Galina Shagieva, Ekaterina Mikheeva, Aigul Galembikova, Pavel Dunaev, Dmitry Kudlay, Sergei Boichuk, Pavel Kopnin

**Affiliations:** 1A.N. Belozersky Institute of Physico-Chemical Biology, Lomonosov Moscow State University, Moscow 119991, Russia; vdugina@iname.com (V.D.); shagievags@my.msu.ru (G.S.); 2Biological Faculty, M.V. Lomonosov Moscow State University, Moscow 119991, Russia; 3Scientific Research Institute of Carcinogenesis, N. N. Blokhin National Medical Research Center of Oncology, Moscow 115522, Russia; mvnovikova94@mail.ru (M.V.); nkhromova@gmail.com (N.K.); vinokourova@mail.ru (S.V.); 4Department of Pathology, Kazan State Medical University, Kazan 420012, Russia; miheeva.1973@bk.ru (E.M.); ailuk000@mail.ru (A.G.); dunaevpavel@mail.ru (P.D.); boichuksergei@mail.ru (S.B.); 5Department of Pharmacology, The I. M. Sechenov First Moscow State Medical University (The Sechenov University), Moscow 119991, Russia; d624254@gmail.com; 6Department of Pharmacognosy and Industrial Pharmacy, Lomonosov Moscow State University, Moscow 119992, Russia; 7Department of Radiotherapy and Radiology, Russian Medical Academy of Continuous Professional Education, Moscow 119454, Russia; 8“Biomarker” Research Laboratory, Institute of Fundamental Medicine and Biology, Kazan Federal University, Kazan 420008, Russia

**Keywords:** breast cancer, actin isoforms, EMT, taxol resistance, microtubule reorganisation, pro-metastatic changes

## Abstract

The widespread occurrence of breast cancer and its propensity to develop drug resistance highlight the need for a comprehensive understanding of the molecular mechanisms involved. This study investigates the intricate pathways associated with secondary resistance to taxol in triple-negative breast cancer (TNBC) cells, with a particular focus on the changes observed in the cytoplasmic actin isoforms. By studying a taxol-resistant TNBC cell line, we revealed a shift between actin isoforms towards γ-actin predominance, accompanied by increased motility and invasive properties. This was associated with altered tubulin isotype expression and reorganisation of the microtubule system. In addition, we have shown that taxol-resistant TNBC cells underwent epithelial-to-mesenchymal transition (EMT), as evidenced by Twist1-mediated downregulation of E-cadherin expression and increased nuclear translocation of β-catenin. The RNA profiling analysis revealed that taxol-resistant cells exhibited significantly increased positive regulation of cell migration, hormone response, cell–substrate adhesion, and actin filament-based processes compared with naïve TNBC cells. Notably, taxol-resistant cells exhibited a reduced proliferation rate, which was associated with an increased invasiveness in vitro and in vivo, revealing a complex interplay between proliferative and metastatic potential. This study suggests that prolonged exposure to taxol and acquisition of taxol resistance may lead to pro-metastatic changes in the TNBC cell line.

## 1. Introduction

Breast cancer is among the most common human malignancies. Current chemotherapy regimens for breast cancer are based on combinations of anthracyclines (such as doxorubicin, epirubicin, and daunomycin) and/or taxanes (such as taxol/paclitaxel (PTX) and docetaxel) [1]. Unfortunately, classical effective therapeutic strategies often lead to the acquisition of drug resistance in human tumours [2], which is one of the main reasons for relapse and metastasis. Multiple and partially overlapping mechanisms are known to underlie the chemoresistance of breast cancer. These include alterations in drug absorption, transport and efflux, transporter proteins, DNA repair, the tumour microenvironment, and activation of the epithelial–mesenchymal transition (EMT) pathway [3,4].

Breast tumours lacking oestrogen receptor (ER), progesterone receptor (PR), and human epidermal growth factor receptor 2 (HER2) are considered triple-negative breast cancer (TNBC). TNBC accounts for approximately 15–20% of all breast cancers and is known to be associated with aggressive clinical behaviour and poor survival [5]. Our previous findings show that the ratio of cytoplasmic actin isoforms has a significant impact on the phenotype and karyotype of the TNBC cell line MDA-MB-231. Furthermore, β-actin depletion and reciprocal γ-actin increase promoted chromosomal instability in MDA-MB-231 cells [6]. It has been shown that γ-actin affects microtubule dynamics: the predominance of γ-actin causes microtubule instability [7], while partial depletion of γ-actin reduces microtubule dynamics in the interphase cells [8]. Given that the taxol is a well-known microtubule-stabilising agent, a shift in the balance of cytoplasmic actins might be responsible for the development of drug resistance to microtubule-targeting agents in tumour cells.

In this study, we compared the characteristics of the naïve and taxol-resistant TNBC HCC1806 cell lines previously established in Dr Boichuk’s laboratory [9]. A notable shift in cytoplasmic actin expression towards γ-actin was observed in the taxol-resistant HCC1806-TaxR subline compared with parental HCC1806 cells. Concurrently, we detected increased motility and invasive properties in the taxol-resistant subline, accompanied by reduced proliferation rates both in vitro and in vivo. Additionally, our findings revealed significant activation of the EMT-related signalling pathways, thereby suggesting that exposure to taxol and the development of taxol resistance activate EMT in breast cancer cells.

## 2. Results

### 2.1. Characterisation of the Taxol-Resistant HCC1806 Cell Line

The drug sensitivity of the taxol-resistant HCC1806 subline was determined using a 72 h cytotoxicity assay. Naïve HCC1806 cells were highly sensitive to taxol with an IC50 of 0.25 nM. The resistant HCC1806-TaxR subline, continuously maintained in the presence of 10 nM taxol, displayed a high level of resistance to the selecting drug compared with parental HCC1806 cells (Figure 1a). The fold resistance was determined as the ratio of the IC50 of the resistant cells to the IC50 of the parental HCC1806 cells. As shown in Figure 1b, HCC1806-TaxR cells exhibited a 20-fold decrease in sensitivity to taxol (Figure 1b). A significant increase in P-glycoprotein (MDR1) expression was also observed in the taxol-resistant HCC1806-TaxR cells (Figure 1c), providing a potential molecular mechanism for their resistance to taxol. Notably, the expression of MDR1 protein was not detectable in the parental HCC1806 cells (Figure 1c). Based on transcriptome analysis, *ABCB1* (MDR1) mRNA expression was found to be 1500-fold higher in resistant HCC1806-TaxR cells than in parental cells. In addition, HCC1806-TaxR cells acquired more mesenchymal phenotype in vitro compared with parental HCC1806 cell culture (Figure 1d). Using bioinformatics analysis and RNA profiling data for naïve HCC1806 and HCC1806-TaxR cells, an interaction between cytoplasmic actins, tubulin, and MDR1 was studied. The possibility of protein–protein interactions (PPI) between MDR1 and the cytoplasmic actins was investigated using STRING (https://string-db.org/; accessed on 9 November 2023). The PPI between MDR1 (*ABCB1*), tubulin and both of the cytoplasmic actins (*ACTB* and *ACTG1*) is shown in Figure 1e.

### 2.2. Proliferative and Invasive Features of Naïve and Taxol-Resistant TNBC Cells

Despite the reduced proliferation capacity, HCC1806-TaxR cells exhibited a more invasive phenotype in vitro compared with parental HCC1806 cells (Figure 2). HCC1806-TaxR cells demonstrated the following characteristics:
Consistent with the level of resistance (Figure 1a), we observed high proliferative activity of HCC1806-TaxR cells cultured in the presence of taxol (paclitaxel, PTX 0.1 and 1 μM), thereby revealing loss of sensitivity of these cancer cells to this particular chemotherapeutic agent. Moreover, proliferative activity of HCC1806-TaxR cells cultured in the presence of taxol was even higher when compared with non-treated HCC1806-TaxR (Figure 2a). As expected, the proliferative activity of taxol-treated naïve HCC1806 cells was significantly reduced, as shown in Figure 2a. Real-time monitoring of cellular proliferation capacity by the iCELLigence system showed that the growth of HCC1806-TaxR cells was significantly lower than that of parental HCC1806 cells (Figure 2b) (10^4^ initial cells/well, 96 h of incubation). By day 7 of incubation, the number of viable HCC1806-TaxR cells was 5-fold lower compared with parental cells, according to phase contrast microscopy (Figure 2c);HCC1806-TaxR cells had an increased ability to form colonies under adhesive conditions (10^4^ cells/dish, 14 days of incubation) compared to parental cells (Figure 2d,e);The data from the Matrigel migration/experimental invasion assay (250 × 10^3^ cells/well, 12 h of incubation, 8 µm filter pore size) demonstrated that HCC1806-TaxR cells exhibited a more motile and invasive phenotype than HCC1806-naïve cells (Figure 2f,g).


**Figure 2 ijms-25-04530-f002:**
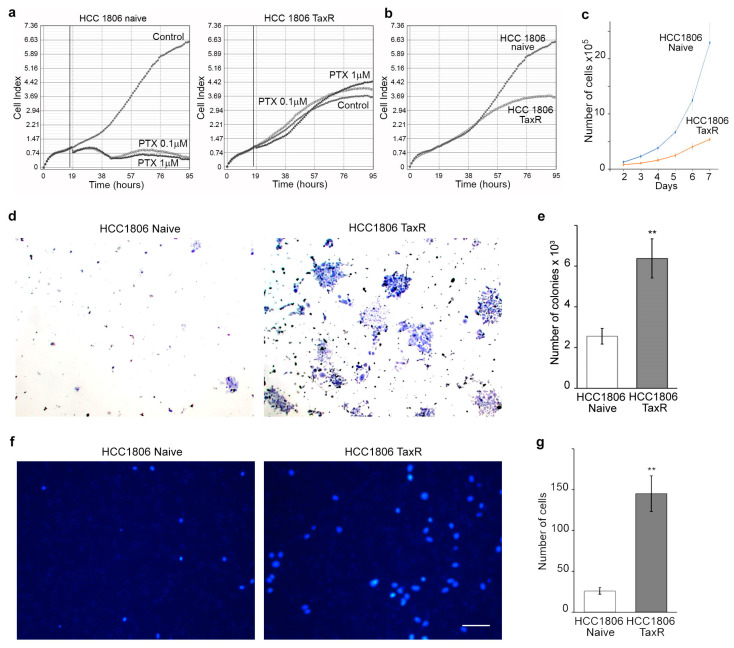
Proliferative and invasive features of naïve and taxol-resistant HCC1806 cells. (**a**) The growth curves of HCC1806-naïve and HCC1806-TaxR cells treated with taxol (paclitaxel, PTX, 0.1 and 1 μM). The vertical lines represent a time-point when taxol was added to the cell cultures; (**b**) comparative growth kinetics of HCC1806-naïve and HCC1806-TaxR cells. iCELLigence system (**a**,**b**); (**c**) HCC1806-naïve and HCC1806-TaxR cell proliferation rate in culture, 2–7 d, phase contrast microscopy; (**d**,**e**) formation of colonies under adherent conditions (10^4^ cells/dish, 14 d) in HCC1806-naïve and HCC1806-TaxR cells, light field microscopy, Mann–Whitney U test; (**f**,**g**) matrigel migration/experimental invasion assay of HCC1806-naïve and HCC1806-TaxR cells (250 × 10^3^ cells/well, 12 h, 8 µm filter pore size), DNA-blue, fluorescent microscopy, scale bar 50 μm, Mann–Whitney U test. Asterisks indicate *p*-values < 0.01 (**) for all panels.

### 2.3. Reorganisation of the Cytoskeleton and the Intercellular Junctions in the Naïve and Taxol-Resistant HCC1806 Cells

According to the literature data, prolonged exposure of cells to taxol leads to microtubule reorganisation. Immunofluorescence analysis revealed that the microtubule system in HCC1806-TaxR cells had a more pronounced radial organisation, corresponding to planar polarity, compared with the epithelial-like morphology of naïve HCC1806 cells (Figure 3a). We also performed Western blotting analysis to examine the expression of total α-tubulin, total β-tubulin, and βI- and βIII-tubulin isotypes in both cell cultures. Quantitative analysis showed that the relative level of α-tubulin was approximately 2-fold higher in HCC1806-TaxR cells than in HCC1806-naïve cells. The expression of other tubulin isotypes was also altered in HCC1806-TaxR cells (Figure 3b).

We have previously shown that microtubule organisation might be a consequence of the changes in the ratio between the cytoplasmic β and γ isoforms of actin [7] and that the predominance of γ-actin correlates with increases in the tumourigenic potential of cancer cells [10]. Western blotting and immunofluorescence analysis of actin isoforms showed that the balance of cytoplasmic actins shifts towards γ-actin predominance in resistant HCC1806-TaxR cells (Figure 4a,b).

E-cadherin, β-catenin, and cytoplasmic β-actin are involved in formation of the intercellular adhesion junctions [11]. These junctions become rearranged during progression of the epithelial malignancies, thereby allowing cancer cells to invade and metastasise. This process is known as the epithelial-to-mesenchymal transition (EMT). Multiple mechanisms can suppress E-cadherin (*CDH1*) during EMT. It has been demonstrated that the E-cadherin gene was downregulated in HCC1806-TaxR cells, likely due to the increased expression of *TWIST1*, a transcription factor and negative regulator of *CDH1* (Figure 4c). TCF/Lef transactivation was upregulated in HCC1806-TaxR cells compared with naïve HCC1806 cells (Figure 4d). Immunofluorescence analysis has confirmed a decreased expression of E-cadherin and revealed suppression of β-catenin in resistant TNBC cells compared to naïve cells. It is important to note that β-catenin staining was detected solely in the nucleus, along with its disappearance from intercellular contacts in resistant cells (Figure 4e). Figure 4f shows clear changes in the expression of EMT markers in HCC1806-TaxR cells compared with their parental counterparts. Specifically, HCC1806-TaxR cells exhibited decreased expression of E-cadherin, β-catenin, and claudin-1, while the expression of N-cadherin and vimentin was increased. Thus, Western blot analysis data were consistent with the immunofluorescence data, as shown in Figure 4e.

### 2.4. Tumour Growth of HCC1806-Naïve and HCC1806-TaxR Cells In Vivo

We also observed modulation of oncogenic properties in vivo in HCC1806-TaxR cells. The minimum inoculation dose for HCC1806-TaxR cells reduced by 2–3 times compared with HCC1806-naïve cells (Figure 5a). Despite this, HCC1806-TaxR cells resulted in decreased tumour growth rates in vivo in xenografts after subcutaneous injection in the BALB/c athymic mice model (Figure 5b), which is consistent with the obtained in vitro data (reduced cell proliferation rates). The HCC1806-TaxR tumour volume was five times smaller 28 days after inoculation (Figure 5c). The histology of both HCC1806-naïve and HCC1806-TaxR xenograft sections of primary tumours and of lymph nodes was examined (Figure 5d). However, it should be noted that HCC1806-TaxR cells had an increased metastatic potential. The percentage of mice with detected metastases in lymph nodes was not increased, whereas the percentage of mice with detected metastases in lung and liver was increased for HCC1806-TaxR compared with HCC1806-naïve cells (Figure 5e). Positive staining for human keratins using antibodies against CK7 on xenograft sections confirmed the human origin of metastatic cells in mouse lymph nodes. The quantitative difference between HCC1806-naïve and HCC1806-TaxR cells was evident in the density of metastatic cells in lymph nodes (Figure 5d,f).

### 2.5. Transcriptome Analysis and Identification of Differentially Expressed Genes Potentially Involved in the Development of the Taxol Resistance Phenotype of the HCC1806 Subline

The analysis of differentially expressed genes (DEGs) between the naïve and taxol-resistant TNBC sublines resulted in 36,596 commonly expressed genes, of which 2048 were upregulated and 2093 were downregulated (Figure 6a) in the HCC1806-TaxR cells. The Metascape functional enrichment analysis revealed that the upregulated DEGs between HCC1806-TaxR and HCC1806-naïve cells had significantly enhanced positive regulation of cell migration, hormone response, cell–substrate adhesion, actin filament-based processes and cell growth (as shown in Figure 6b). The bioinformatics analysis of potential protein–protein bonding of cytoplasmic actins (*ACTB*/*ACTG1*), tubulin and DEGs in resistant cells revealed interactions with genes whose products are involved in processes of cell motility, regulation of the cytoskeleton and extracellular matrix, and various intracellular signalling pathways (Figure 6c–f).

## 3. Discussion

Breast cancer is the most frequently diagnosed cancer in women worldwide and the second leading cause of cancer-related mortality [12]. The molecular and pathological subtypes of breast cancer are highly heterogeneous [13,14]. This study was conducted using the triple-negative breast cancer (TNBC) cell line HCC1806 and its taxol-resistant counterpart. Taxol, also known as paclitaxel (PTX), was isolated by Drs. Wall and Wani and their colleagues, who also documented its anti-tumour activity [15]. Taxol is a frontline chemotherapeutic agent in the treatment of breast cancer, particularly advanced metastatic disease. However, acquired drug resistance frequently occurs, reducing the effectiveness of chemotherapy in human tumours [16]. Therefore, understanding the molecular mechanisms of taxol resistance is crucial to overcoming it and improving potential therapeutic combinations. The development of resistance in cancer is often multifactorial, with complex interactions between different mechanisms that promote cancer survival.

Taxol resistance in cancer is often associated with overexpression of transporters that block drug accumulation in cancer cells and mutations that prevent the drug from binding to its target [17]. Multidrug resistance (MDR) studies involve the examination of cell cultures that have developed resistance to a specific substance. MDR cells were originally selected for insensitivity to a single hydrophobic drug but acquired resistance to a variety of structurally and functionally unrelated lipophilic agents. We used the taxol-resistant subline of HCC1806 cells, which was continuously maintained in the presence of 10 nM taxol. MDR cells often overproduce a plasma membrane glycoprotein known as P-glycoprotein or MDR1, which is encoded by the *ABCB1* gene. This protein acts as an ATP-dependent drug efflux pump to maintain drug concentrations below cytotoxic levels. In our study, we found that HCC1806-TaxR cells had increased levels of P-glycoprotein and increased expression of the *ABCB1* gene. The *ABCB1* gene was also found to be upregulated in untreated tumours of various types, in human multidrug-resistant cells, and in cells selected with different drugs [18]. P-glycoprotein expression in certain cancers may indicate a more aggressive subpopulation of tumour cells with multiple resistance mechanisms. However, some cell lines resistant to microtubule stabilising agents lack P-glycoprotein expression. Examples include the taxol-resistant human lung carcinoma cell line A549-T12 and the taxol-resistant human breast cancer cell line derived from MDA-MB-231-K20T. It has been suggested that β-tubulin mutations contribute to taxol resistance in these cultures [19,20,21].

The morphology of HCC1806-TaxR cells, compared with naïve HCC1806 cells in culture, indicates a reorganisation from an island-oriented to a more motile and pro-metastatic phenotype. This phenotype may indicate an increased metastatic potential of the cells and activation of EMT in taxol-resistant culture. The levels of junctional proteins E-cadherin, β-catenin, and claudin-1 were significantly decreased in HCC1806-TaxR compared with parental cells. In contrast, the expression of vimentin and N-cadherin increased in HCC1806-TaxR cells, indicating the progression of EMT in this breast cancer cell line that has developed resistance to taxol. Furthermore, we have found that breast cancer cells resistant to taxol were more effective at forming colonies and migrating through filters. These properties were accompanied by increased activation of genes associated with migration. At the same time, we observed a decrease in proliferation rates and suppression of genes related to cell division compared with the paternal cell culture. This shift towards migration was most likely due to the strong activation of EMT-related signalling pathways. EMT is an early step in cancer metastasis in which epithelial cells lose cell–cell junctions and polarisation, acquire motile properties, and become invasive, acquiring the characteristics of mesenchymal cells [22]. EMT enhances the tumourigenic and metastatic potential of cancer cells, as well as increasing their resistance to multiple therapeutic regimens [23]. The bioinformatics analysis for HCC1806-TaxR cells, compared to HCC1806-naïve cells, aligns with our morpho-functional experimental data.

According to a revised TNBC tumour classification, taxol-resistant HCC1806-TaxR sublines display similar expression profiles related to EMT, cell motility, and differentiation as mesenchymal-like and basal-like TNBC subtypes [14]. TNBC subtypes differ significantly in terms of their prognosis and response to chemotherapy, as well as in patterns of recurrence. For instance, mesenchymal-like TNBC subtypes have a tendency to metastasize to the lungs [14]. This could explain the higher percentage of lung metastases composed of HCC1806-TaxR cells detected in the BALB/c athymic mice model.

Injections and inoculations of tumour cells are commonly used to assess the malignant potential of cancer cells. The minimum inoculation dose for taxol-resistant cells was 2–3 times lower than that for naïve cells. However, in the BALB/c athymic mouse model, HCC1806-TaxR cells formed smaller tumours. This is consistent with the reduced cell proliferation rates observed in vitro. Another taxol-resistant TNBC cell line also showed a reduced proliferation rate compared with the parental cell line [24]. Slow-cycling may contribute to chemotherapy resistance and other tumour-promoting properties [25]. Immunohistochemical (IHC) analysis of human keratin 7 (K7) expression in mouse lymph nodes in this model revealed abundant K7 staining only in mice injected with taxol-resistant, but not naïve TNBC cells. It has previously been shown that the proliferation and invasive capabilities of metastatic breast cancer cells can be dissociated and oppositely regulated in vivo. The non-receptor kinase Arg/Abl2 controls the cell’s decision to either “grow” or “go” [26]. Migratory cells extracted from a transgenic mouse mammary tumour exhibited lower proliferation and apoptosis levels compared to the mean of primary tumour cells. As a result, the subpopulation of migrating cells showed increased resistance to chemotherapy compared with the stationary tumour cells [27].

Examination of the cytoplasmic actin profile and β-tubulin isotypes revealed the altered isoform balance in HCC1806-TaxR cells compared with HCC1806-naïve cells. Differential expression of tubulin isotypes, tubulin mutations, and post-translational modifications of tubulin have been associated with tumour aggressiveness and drug resistance [28]. At least eight α- and eight β-tubulin isotypes have been identified in different human cells in varying amounts, and constitutive isotype expression appears to be tissue-specific. Taxol interacts with β-tubulin isotypes in a unique way [29]. Therefore, a patient’s response to the drug may depend, in part, on the isotype content of the tumour cells. The differential expression of β-tubulin isotypes in human cancer cells and their taxol-resistant analogues has been previously documented [28,30]. For example, in the taxol-resistant cell line AT12, βIII- and βIV-tubulins were slightly increased (1.3- and 1.4-fold, respectively) compared to the paternal A549 human lung cancer cell line [31]. We found increased levels of α-tubulin, βI-, and βIII-tubulin isotypes in taxol-resistant HCC1806-TaxR cells. In several cancers, overexpression of βIII-tubulin has been associated with resistance to taxol [28,30,32]. Differences in β-tubulin isotypes may be predictive markers for the development of therapeutic strategies with microtubule interacting agents (MIAs). It has been observed that βIII-tubulin binds the least amount of a tritium-labelled taxol analogue, 2-m-AzTax, compared to other β-tubulin isotypes [33]. Variations in the primary structure of βIII-tubulin compared to βI-tubulin affect the taxol-binding domain [34] and are likely responsible for the ability of this isotype to confer resistance to taxol-induced apoptosis [35]. A significant increase in the expression of βI (3.6-fold), βIII (4.4-fold) and βIVa (7.6-fold) tubulin isotypes was detected in the taxol-resistant samples compared with untreated primary ovarian tumours [30]. However, studies have shown that pre-operative neoadjuvant taxane-containing chemotherapy has a positive response in ER-negative breast cancers with high levels of βIII-tubulin expression [36]. Further investigation is required to determine the role of βIII-tubulin overexpression in taxane-based chemotherapy resistance.

Multiple factors were proposed to have an impact on the tubulin binding. In particular, alterations in the actin cytoskeleton may affect microtubules and can also contribute to drug resistance. It has previously been suggested that there is a link between actin, tubulin, and taxol, and that actin may be involved in resistance to microtubule-interacting agents [28,37,38]. The actin cytoskeleton is composed of different actin isoforms and actin-binding proteins. It has been shown that β-actin has a preferential role in contractile and adhesive structures, whereas γ-actin forms the cortical network necessary for cell shape flexibility and motile activity [10]. The two actin isoforms are differentially involved in the processes of cell division [39,40], regulation of epithelial junctions [11,41], directional motility [7,10], and neoplastic cell transformation [10]. Previously, we observed changes in proliferation, migration, and invasion in various tumour cell cultures as a result of a shift in the ratio of actin isoforms. γ-Actin overexpression/β-actin suppression led to increased motility in the experimental invasion assay in vitro and induced upregulation of some oncogenic features in vivo. A direct interaction between γ-cytoplasmic actin and the microtubule plus-end tracking protein (+TIP) EB1 has been demonstrated in human breast adenocarcinoma MCF7 cells with induced γ-actin predominance [7]. This interaction may help cells to differentially regulate microtubule stability in the actin-rich cortex as opposed to the cell interior [42]. The predominance of γ-actin caused microtubule instability [7], while the partial depletion of γ-actin reduced microtubule dynamics in interphase neuroblastoma cells [8]. Changes in γ-actin have been implicated in tubulin-targeted drug resistance in childhood leukaemia [38]. Therefore, γ-actin may play a crucial role in achieving taxol resistance. A variety of cytoskeletal and cytoskeleton-associated proteins, including galectin-1, 14-3-3σ and phosphorylated statmin, were found to be differentially expressed in drug-resistant cells [31]. The first direct visualisation and assessment of in vivo migration of mammary tumour cells showed that migrating tumour cells exhibited coordinated gene expression changes, including activation of actin polymerisation and myosin contraction [43]. One of the prominently upregulated tumour cell dissemination pathways is the mammalian-enabled (Mena)-cofilin pathway [43,44,45]. Mena is an actin-binding protein involved in the regulation of cofilin-stimulated actin regulation, which determines chemotactic orientation and invasion [46]. Mena isoform expression pattern is a clinically validated prognostic marker for metastasis, predicting metastatic relapse and survival in breast cancer patients [47]. Cofilin1, as we have previously shown, selectively interacts with γ-actin, and there is a γ-actin-dependent regulation of the expression of cofilin1 [10]. The interplay between cofilin and γ-actin takes place at the leading edge of the cell, as detected by proximity ligation assay (PLA), so these interactions could provide a basis for explaining the necessity of γ-actin for migration and invasion.

In addition to drug resistance, the therapy-driven tumour progression has been observed in preclinical and clinical studies [48]. Cytotoxic chemotherapy paradoxically promotes systemic dissemination of cancer cells to secondary sites. Taxol directly binds to and activates Toll-like receptor-4 (TLR4), which is often overexpressed on the surface of breast cancer cells, and induces a number of pro-inflammatory regulators that promote distant metastasis [49,50]. Taxol chemotherapy enhanced pro-metastatic properties in breast cancer models. This process was mediated, at least in part, by tumour-derived extracellular vesicles enriched in annexin-A6 (ANXA6) [51]. Gingis-Velitski and colleagues suggested that distinct categories of drugs stimulate the cytokine storm and therefore mediate cancer cell invasion and dissemination via different mechanisms [52]. Stratifying patients into groups with a high or low risk of metastasis could help reduce overtreatment with chemotherapy in patients with a good prognosis. Breast cancer patients would benefit from additional tests that measure different aspects of tumour biology, such as dissemination. The process of tumour cell evolution under the influence of treatment may be common across tumour types. Initial treatment that induces remission is often followed by invisible tumour progression and relapse, as well as treatment failure due to the development of drug resistance [53]. We propose that combination therapy during tumour regression after taxol treatment is essential to prevent dissemination and tumour progression.

Chemotherapy is a frontier therapy that is essential for cancer control, but unfortunately it can be associated with adverse effects and drug resistance in some patients. Taxol treatment has been shown to induce critical functional regulations in the tumour’s microenvironment to promote cancer cell development and progression following chemotherapy [48]. We have shown that even in the absence of surrounding tissue, exposure to taxol and taxol resistance stimulates pro-metastatic changes in TNBC cell line.

## 4. Materials and Methods

### 4.1. Cell Culture

Parental (i.e., naïve) TNBC HCC1806 cell line was purchased from American Tissue Culture Collection (ATCC, CRL-2335). The HCC1806-TaxR subline, which is resistant to taxol, was established in Dr. Boichuk’s laboratory through stepwise treatment with increasing concentrations of taxol, as previously described [9]. The cell lines were cultivated in complete RPMI-1640 culture medium (Paneco, Moscow, Russia) with 10% foetal bovine serum (Gibco; Thermo Fisher Scientific, Waltham, MA, USA) and antibiotics (e.g., 50 U/mL penicillin and 50 µg/mL streptomycin). The cells were maintained at 37 °C in a humidified atmosphere of 5% CO_2_ in the incubator (LamSystems, Miass, Russia).

### 4.2. Cytotoxicity Assay/Cellular Survival MTS-Based Assay

To determine the IC50 values of taxol/paclitaxel for both naive and resistant HCC1806 cells, cancer cells were seeded into 6-well flat-bottomed plates (Corning, Corning, NY, USA) and allowed to attach and grow for 24 h. Subsequently, the cells were treated with varying concentrations of taxol/paclitaxel or DMSO (as control) for 48–72 h. Finally, the MTS reagent (Promega, Madison, WI, USA) was added to the cell culture for 1 h to assess live cell numbers. Optical density (OD) measurements were taken at 492 nm using a MultiScan FC plate reader (Thermo Fisher Scientific, USA). The IC50 values were determined as the concentration of the compound required to inhibit cellular growth by 50% within 24–48 h. The experiments were performed in three replicates and the data were normalised to DMSO-treated cells.

### 4.3. Real-Time Monitoring of Cell Proliferation

The proliferation capacity of cancer cells was examined in a real-time iCELLigence system (ACEA Biosciences, Santa Clara, CA, USA). For this purpose, cells were seeded in electronic microtiter plates (E-Plate; Roche Diagnostics, Munich, Germany) and further cultured for 24 h to obtain their initial growth. Next, cancer cells were treated with 0.1 or 1 µM of taxol/paclitaxel for the following 72 h. DMSO was used as a negative control for cancer cell growth. Cell index (CI) was measured for every 30 min until the endpoint of the experiment. Normalised cell index (NCI) values were assessed by using the RTCA Data Analysis Software version 1.0 (ACEA Biosciences, USA).

### 4.4. Cell Proliferation Rate

Cells were seeded in Petri dishes at 5 × 10^4^ cells/dish. The number of cells was determined with a haemocytometer every 24 h.

### 4.5. Colony Formation Assay

The cells (10^4^) were seeded onto 10 cm Petri dishes and incubated for 14 days. Afterward, they were fixed with 70% ethanol and stained with Giemsa dye (PanEco, Russia). The Total Lab ver. 2.0 software, Colony Counter module (Nonlinear Dynamics, Newcastle upon Tyne, UK), was used to determine the number of colonies and their sizes.

### 4.6. Boyden Chamber Cell Migration/Experimental Invasion Assay

Firstly, 2.5 × 10^5^ cells/well were seeded in the plates with the integrated Boyden chambers (filter pore diameter, 8 μm, Matrigel coated). After 12 h of incubation, the cells remaining on the filter upper surface were removed and the migrated cells were fixed with methanol and stained with 4′,6-diamidino-2-phenylindole (DAPI) (D9542, Sigma-Aldrich, St. Louis, MO, USA). The migration activity was determined by counting the cells on the filter bottom surface using an Axiovert 200 microscope (Carl Zeiss, Jena, Germany).

### 4.7. Immunofluorescence Microscopy

The cells were cultured on glass coverslips washed with pre-warmed DMEM containing 20 mM 4-(2-hydroxyethyl)-1-piperazineethanesulfonic acid (HEPES) at 37 °C. Subsequently, the cells were fixed with 2% paraformaldehyde (PFA) in serum-free DMEM (with 20 mM HEPES) for 10 min and extracted with cold methanol (MeOH) for 5 min at −20 °C for actin, E-cadherin, and β-catenin antibody staining. For tubulin antibody staining, we used fixation with cold MeOH at −20 °C. Immunofluorescence was observed using an Axioplan microscope with 40×/0.75 and 100×/1.3 Plan-Neofluar lenses (Carl Zeiss).

### 4.8. Western Blotting

To prepare whole-cell extracts, we scraped the cells from culture dishes into ice-cold RIPA buffer (25 mM Tris-HCl pH 7.6, 150 mM NaCl, 5 mM EDTA, 1% NP-40, 1% sodium deoxycholate, and 0.1% SDS), containing protease and phosphatase inhibitors. We incubated the lysates at 4 °C for 1 h in RIPA buffer and clarified them via centrifugation at 13,000 rpm for 30 min at 4 °C. The measurement of the protein concentrations in cell lysates was performed using the Bradford assay. The samples with 30 µg of protein were loaded onto 4 to 12% Bis-Tris or 3 to 8% Tris-acetate NuPAGE gels (Invitrogen, Waltham, MA, USA), and then transferred onto nitrocellulose membranes (Bio-Rad, Hercules, CA, USA). The membranes were blocked with SuperBlock Blocking Buffer (Thermo Scientific, USA) for 30 min at room temperature before being stained overnight with specific antibodies at 4 °C. The membranes were washed three times with ice-cold PBS. Then, the secondary antibodies were applied, followed by another wash with PBS. Finally, the bands were detected using enhanced chemiluminescence (Western Lightning Plus-ECL reagent, Perkin Elmer, Waltham, MA, USA). The densitometric analysis of the Western blotting images was performed using NIH ImageJ software (version 1.49) (Bethesda, Rockville, MD, USA) and normalised to the total actin levels. We estimated the protein amount by analysing the results of at least three independent experiments.

### 4.9. Antibodies

The following primary antibodies were used: mouse monoclonal antibodies to α-tubulin (clone DM1A, IgG1, Invitrogen); β-tubulin (clone TBN06, Invitrogen); h-β-I-tubulin (IgG2b, R&D Systems, Minneapolis, MN, USA); h-β-III-tubulin (IgG2a, R&D Systems); pan-actin (clone C4, Cell Signaling Technology Inc., Danvers, MA, USA); β-actin (IgG1, MCA5775GA, AbD Serotec, Kidlington, UK); γ-actin (IgG2b, (MCA5776GA, AbD Serotec); β-catenin (IgG1, DAKO, Glostrup, Denmark); E-cadherin (IgG2a, BD Transduction); MDR-1 (Santa Cruz Biotechnology, Dallas, TX, USA); N-cadherin (Cell Signaling Technology Inc., Danvers, MA, USA); claudin-1 (Cell Signaling Technology Inc.); and vimentin (V9, DAKO).

The following secondary antibodies were used: AlexaFluor488-, Red-X, AlexaFluor594-conjugated goat anti-mouse IgG or specific to IgG1, IgG2b, IgG2a, absorbed to other IgG isotypes, and goat anti-rabbit IgG (Jackson ImmunoResearch Laboratories Inc., West Grove, PA, USA). DAPI (D9542, Sigma-Aldrich) was applied for DNA (nuclear) staining. HRP-conjugated secondary antibodies were used for Western blot analysis: anti-mouse IgG and anti-rabbit IgG (Santa Cruz Biotechnology).

### 4.10. RNA Sequencing and Transcriptome Analysis

For this study, we obtained two groups of samples for NGS sequencing: the parental TNBC HCC1806 cell line (i.e., HCC1806-naïve) and the taxol-resistant HCC1806 subline, HCC1806-TaxR. Each group was analysed in three replications, with each one representing an independent experiment. Total RNA was isolated from cell line samples using the QIAGEN RNeasy Kit according to the kit protocol (Qiagen, Redwood City, CA, USA). The concentration of total RNA was measured using the Agilent RNA 6000 Nano Qubit RNA Assay Kit (Thermo Fisher, USA). The enrichment of mRNAs and preparation of libraries for sequencing were performed using the KAPA RNA Hyper RiboErase Kit (KAPA Biosystem, Wilmington, MA, USA). The RNA concentrations in the resulting libraries were determined and quality assessments were carried out using the Qubit RNA HS Assay Kit (Life Technologies, Carlsbad, CA, USA) and the Agilent Tapestation instrument (Agilent, Santa Clara, CA, USA). Sequencing of the obtained library was performed using Illumina NextSeq 550 equipment with a read length of 75 bp, with at least 20 million raw reads per sample. For each sample, the resulting RNA sequencing reads were aligned with the H. sapiens UCSC hg38 reference genome (GRCh38). The number of reads per gene values were then normalised using the DESeq2 software package, Version 1.38.1 (https://bioconductor.org/packages/release/bioc/html/DESeq2.html (accessed on 17 April 2024)) and the activation levels of different molecular pathways were calculated based on the normalised gene expression values.

The identification of DEGs and the visualisation of the resulting data were carried out using The SRplot, a free online data analysis platform http://www.bioinformatics.com.cn/srplot (accessed on 9 November 2023). Gene Ontology annotation of the DEGs was performed using Metascape software. The possibility of protein–protein interactions between different sets of DEGs from RNA sequencing analysis with the cytoplasmic actin isoforms and tubulin were investigated using the STRING https://string-db.org/ (accessed on 9 November 2023).

Quantitative real-time PCR (qRT-PCR) with Bio-Rad CFX96 Real-Time System thermocycler (Bio-Rad, Hercules, CA, USA) and SYBR Green I reaction mix (Evrogen, Moscow, Russia) was performed to verify the results of transcriptome analysis. We calculated the Ct values for the target and reference (TUBA1A) genes with Bio-Rad CFX Manager software Version 1.1 (Bio-Rad, USA). We estimated the relative expression of the genes using the 2-ΔΔCt method. The following primers were used (5′→3′): forward—GTCTGTAGGAAGGCACAGCC, reverse—TGCAACGTCGTTACGAGTCA for *CDH1*; forward—CACGAGCGGCTCAGCTACGC, reverse—CGCTGCCCGTCTGGGAATCAC for *TWIST1*; forward—GTTGGTCTGGAATTCTGTCAG, reverse—AAGAAGTCCAAGCTGGAGTTC for *TUBA1A*. The primers were synthesised by Evrogen, Russia.

### 4.11. Luciferase Assay

The activity of the Wnt/beta-catenin signalling pathway in HCC1806 cells was analysed using the Cignal TCF/LEF Reporter Assay Kit (Qiagen, Hilden, Germany) in the Steady-Glo Luciferase Assay System (Promega, Madison, WI, USA). After interaction with the luminogenic substrate, luminescence was detected using a 20/20n Luminometer (Fisher Scientific International, Hampton, NH, USA). The obtained values were normalised to the protein concentration in the samples determined by the Bradford method using the Quick Start Bradford 1× Dye Reagent (Bio-Rad, USA).

### 4.12. BALB/c Athymic Mice Assay

The experiments were conducted on female immunodeficient athymic BALB/c nu/nu mice aged 7–8 weeks and weighing 15–20 g. Each experimental group consisted of 8 to 10 animals. To determine the minimum inoculation cell dose, animals were subcutaneously injected with suspensions containing cell concentrations in the range of 0.25–2.5 × 10^6^ in 100 μL of sterile saline solution. On the 14th day after injection, the presence of a xenograft was determined by palpation.

Cell suspension (2.5 × 10^6^ cells in 100 μL) in sterile saline solution was injected subcutaneously to determine the xenograft growth rates. Linear xenograft parameters (length and width) were measured with a calliper every 4 days for 4 weeks. The xenograft volume was calculated using the formula width^2^ × length × 0.5, where width ≤ length, and the dependence of the xenograft size on days after inoculation was plotted. The cases of metastasis were counted manually.

After the experiments reached their endpoints, the animals were sacrificed, and the tumours and regional lymph nodes were excised and subjected to a histopathologic examination. Formalin-fixed, paraffin-embedded (FFPE) tissues (tumours and lymphatic nodes) were sectioned at 4 μM for haematoxylin and eosin (H&E) stain. The lymphatic nodes were also subjected to IHC-staining using mouse monoclonal antibody to cytokeratin 7 (clone OV-TL 12/30, Cell MarqueTM, Sigma-Aldrich, USA) after antigen demasking in citrate buffer, pH6 for 20 min in Pretreatment Module™ Deparaffinization and Heat-Induced Epitope Retrieval (Thermo Scientific) over 90 °C. The images were captured using ScanScope XT (Aperio Technologies Inc., Vista, CA, USA).

The research was conducted in accordance with ethical standards for the treatment of animals, as adopted by the European Convention for the Protection of Vertebrate Animals used for Experimental and Other Scientific Purposes. The protocol for the animal study was approved by the Ethical Committee of N.N. Blokhin National Medical Research Center of Oncology (decision 05p-17/05/2023).

## 5. Conclusions

Our investigation revealed a significant shift in the balance of cytoplasmic actin isoforms towards γ-actin predominance in triple-negative breast cancer (TNBC) cells that had acquired taxol resistance. The prevalence of γ-actin in taxol-resistant cancer cells was accompanied by increased colony formation, motility, and invasiveness both in vitro and in vivo. The alterations were accompanied by the activation of epithelial–mesenchymal transition and changes in tubulin isotype expression, indicating a complex reprogramming of the cytoskeletal architecture. Transcriptome analysis revealed differentially expressed genes that may be involved in the development of taxol resistance in TNBC cells. These genes were found to be associated with cell migration, hormone response, cell–substrate adhesion, and actin filament-based processes. These findings contribute to our understanding of the mechanism behind taxol resistance. The results suggest that a more strategic approach to chemotherapy, possibly in combination with other treatments, may be necessary to address the pro-metastatic changes caused by taxol resistance.

## Figures and Tables

**Figure 1 ijms-25-04530-f001:**
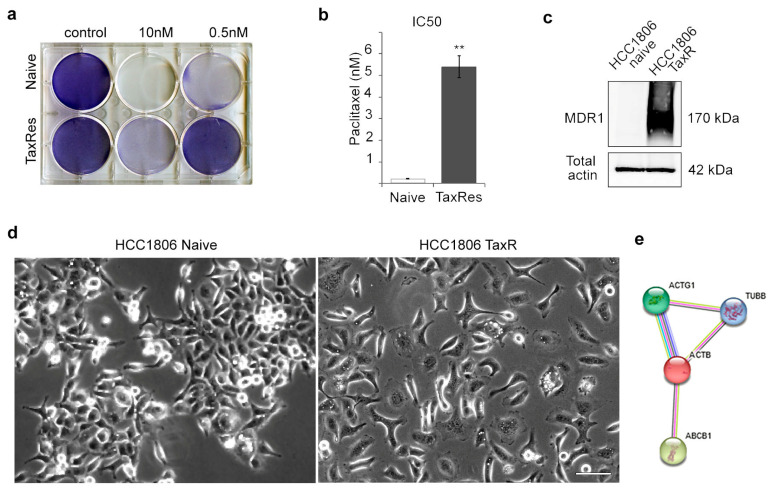
Sensitivity to taxol and morphology of the naïve HCC1806 cell line and the taxol-resistant subline HCC1806-TaxR. (**a**) Cytotoxicity assay to taxol of naïve and taxol-resistant HCC1806 cells (72 h); (**b**) resistance of HCC1806-TaxR and HCC1806 cells to taxol, IC50, cytotoxicity assay, Mann–Whitney U test, asterisks indicate *p*-values < 0.01 (**); (**c**) P-glycoprotein (MDR1) expression in HCC1806 and HCC1806-TaxR cells, Western blotting analysis; (**d**) phenotype of HCC1806-TaxR and HCC1806 cells in vitro, phase contrast microscopy, scale bar 50 μm; (**e**) the interaction map for β- and γ-cytoplasmic actins (*ACTB* and *ACTG1*, respectively), tubulin (*TUBB*), and MDR1 (*ABCB1*), based on bioinformatics analysis and RNA profiling data (https://string-db.org/; accessed on 9 November 2023). Each line indicates one of the different methods that have shown the interaction between proteins, the products of the genes mentioned.

**Figure 3 ijms-25-04530-f003:**
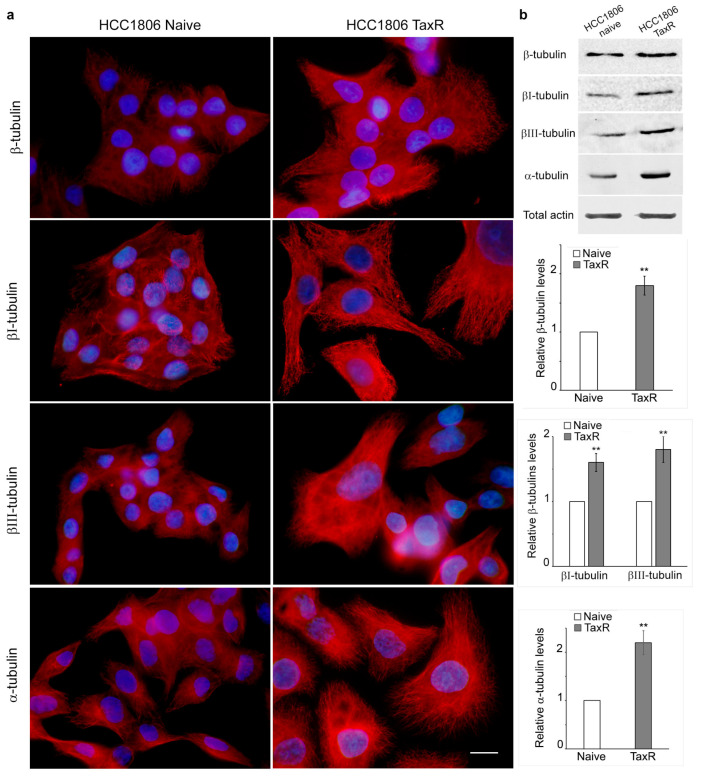
The reorganisation of the microtubule system and tubulin’s levels in the naïve and taxol-resistant HCC1806 cells. (**a**) Microtubules organisation in HCC1806-naïve and HCC1806-TaxR cells, immunofluorescence microscopy, scale bar 10 μm; (**b**) expression of tubulin isotypes in HCC1806-naïve and HCC1806-TaxR cells, Western blotting analysis, Mann–Whitney U test. Asterisks indicate *p*-values < 0.01 (**) for all panels.

**Figure 4 ijms-25-04530-f004:**
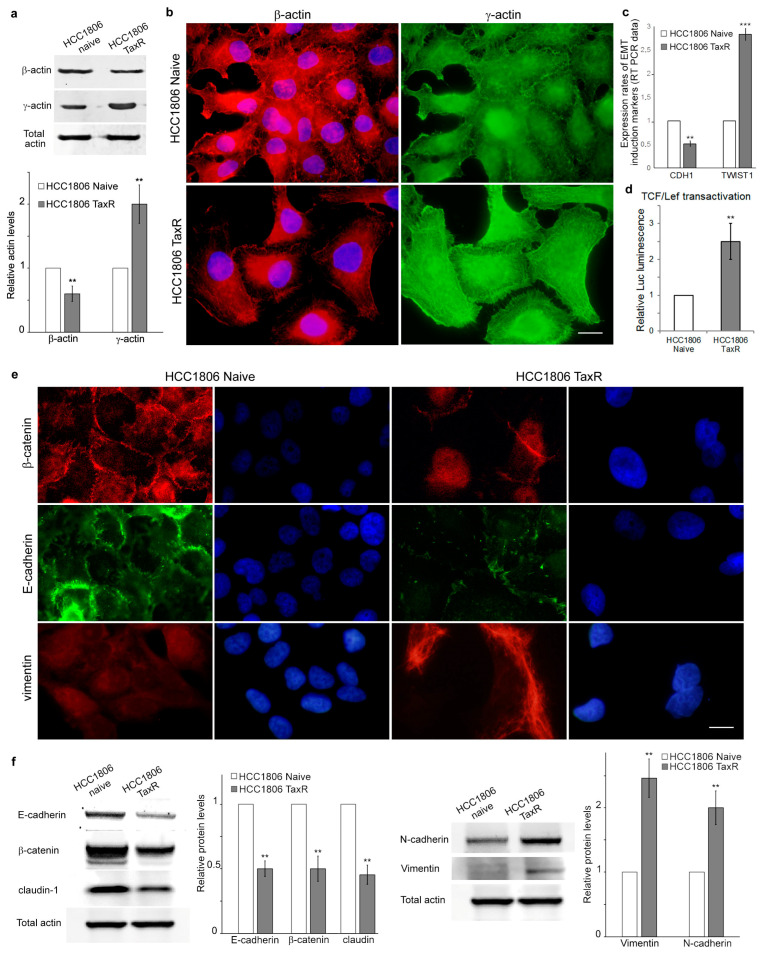
The cytoplasmic actins and adhesion junction proteins in the naïve and taxol-resistant HCC1806 cells. (**a**) Western blot analysis of cytoplasmic actins in HCC1806-naïve and HCC1806-TaxR cells; (**b**) immunofluorescence microscopy of cytoplasmic actins in HCC1806 and HCC1806-TaxR cells, red—cytoplasmic β-actin, green—cytoplasmic γ-actin, blue—DNA; (**c**) comparison of the expression of *CDH1* and *TWIST1* in HCC1806-naïve and HCC1806-TaxR cells, RT PCR analysis; (**d**) TCF/Lef transactivation, Luciferase luminescence analysis; (**e**) immunofluorescence microscopy of E-cadherin, β-catenin and vimentin in HCC1806-naïve and HCC1806-TaxR cells, red—β-catenin, green—E-cadherin, blue—DNA. Scale bars 10 μm; (**f**) Western blot analysis of EMT-related proteins E-Cadherin, β-Catenin, claudin-1 (left), N-Cadherin, vimentin (right) in HCC1806-naïve and HCC1806-TaxR cells. Mann–Whitney U test in (**a**–**d**). Asterisks indicate *p*-values < 0.01 (**) or <0.001 (***) for all panels.

**Figure 5 ijms-25-04530-f005:**
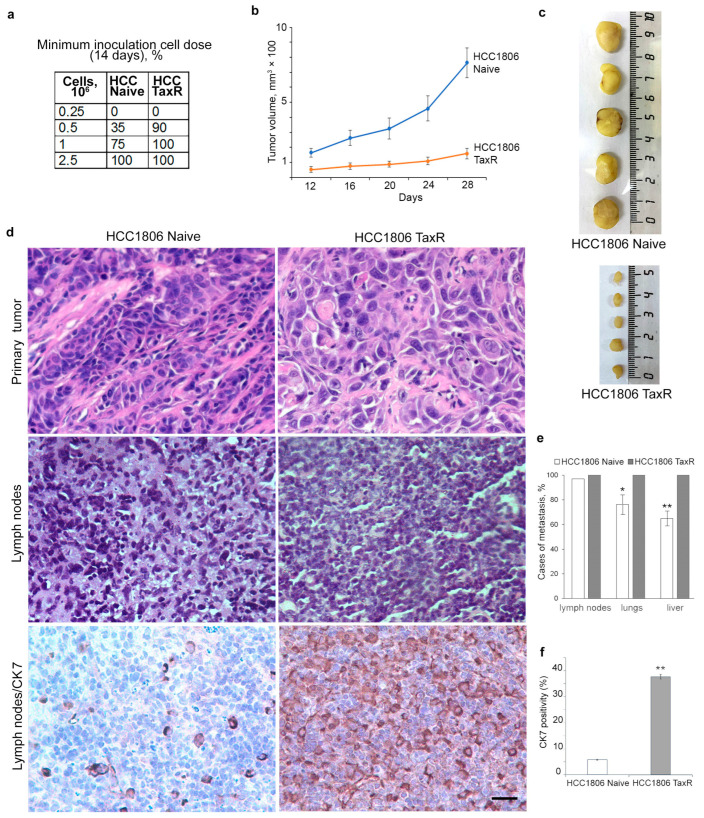
Tumour growth rates of the naïve HCC1806 and HCC1806-TaxR cells in vivo in xenografts. (**a**) The minimum inoculation dose for HCC1806-TaxR and HCC1806-naïve cells; (**b**,**c**) tumour growth in vivo in xenografts after subcutaneous injection, BALB/c athymic mice model; (**d**) histological examination of HCC1806-naïve and HCC1806-TaxR cells and immunohistochemical (IHC) staining for human keratin CK7 in mouse lymph node sections (bottom panel), scale bar 50 μm; (**e**) cases of metastasis for HCC1806-TaxR and HCC1806-naïve xenografts (percentage of mice with detected metastases in different organs), Mann–Whitney U test; (**f**) the density of HCC1806-naïve and HCC1806-TaxR metastatic cells in the mice lymph nodes, IHC analysis for human keratin CK7, Mann–Whitney U test. Asterisks indicate *p*-values < 0.05 (*) or <0.01 (**) for all panels.

**Figure 6 ijms-25-04530-f006:**
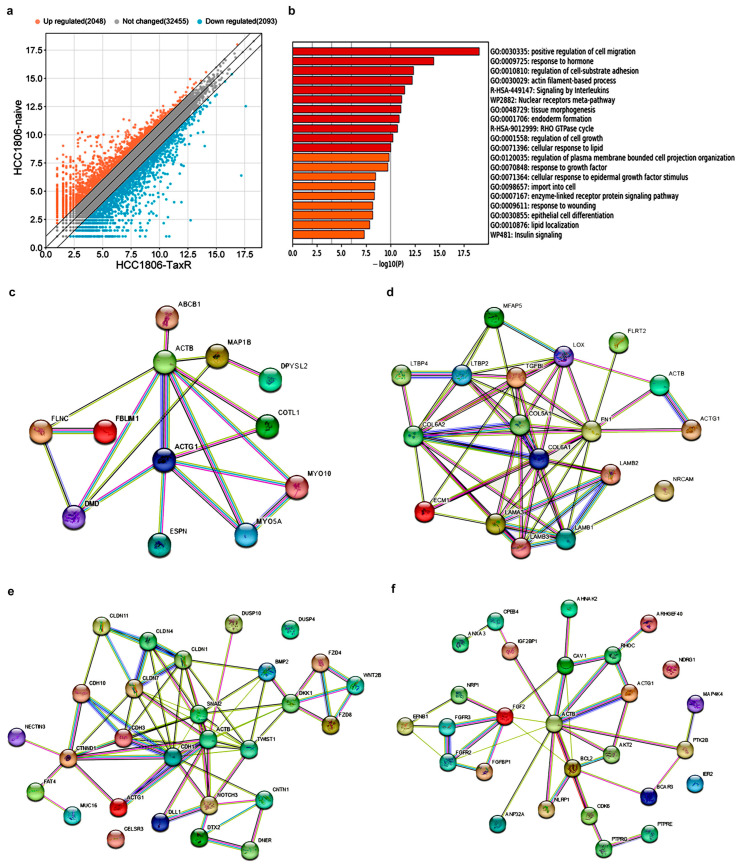
Bioinformatics analysis and visualisation of RNA profiling data for HCC1806-naïve and HCC1806-TaxR cells. (**a**) Scatter plot of DEGs in the naïve HCC1806 and HCC1806-TaxR cells. The X–Y axis represents the log 2-transformed gene expression level, the orange colour represents upregulated genes, the blue colour represents downregulated genes, and the grey colour represents non-DEGs. The scatter plot was generated using https://www.bioinformatics.com.cn/en accessed on 17 April 2024; (**b**) gene ontology enrichment analysis of upregulated genes in HCC1806-TaxR cells compared with naïve HCC1806. A heatmap of enriched terms across input differentially upregulated gene lists, coloured by *p*-values, Metascape software, https://metascape.org/gp/index.html#/main/step1 accessed on 9 November 2023; (**c**–**f**) The interaction “cell motility” maps for EMT (**c**), actin/tubulin cytoskeleton (**d**), extracellular matrix—ECM (**e**) and cell signalling (**f**) with *ACTB*/*ACTG1*. Each line indicates one of the different methods that have shown the interaction between proteins, the products of the genes mentioned.

## Data Availability

The original contributions presented in the study are included in the article; further inquiries can be directed to the corresponding author.

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
