# Peer review of "Imbalance between Actin Isoforms Contributes to Tumour Progression in Taxol-Resistant Triple-Negative Breast Cancer Cells"

_ijms, 2024, doi:10.3390/ijms25084530_

Round 1
Reviewer 1 Report
Comments and Suggestions for Authors
- Authors must clarify if HCC1806-TaxR cells demonstrate reduced or higher proliferation when compared to the naïve ones. On 2.2 Proliferative and invasive features of naïve and taxol-resistant TNBC cells, paragraph 1, line 106, the authors affirm a "reduced proliferation capacity" of HCC1806-TaxR cells. Then, lines 112 and 113 state that resistant cells exhibited high proliferative activity compared to untreated control;
- Cytotoxicity assay and IC50 determination method are not described in this article;
- Line 123: use scientific notation and state the unit (cells/mL, cells/well etc); Lines 531, 533: update scientific notation.
- Regarding figures 3 and 4, HCC1806-TaxR cells appears to be bigger (size and nuclei) than HCC1806 naïve, while figure 1D shows cells of similar size. The authors must build panels 3a, 4b and 4e in the same magnifying glass, preferably indicating the structural changes with arrows;
- Authors must update the bibliography. More than 70% of the references are prior to 2016.
Author Response
Dear Reviewer,
We sincerely thank you for providing us with the opportunity to improve our manuscript. We have carefully considered all of your suggestions and comments and have made the necessary revisions to improve the presentation of our data. Thank you for your work on our manuscript.
Reviewer: Authors must clarify if HCC1806-TaxR cells demonstrate reduced or higher proliferation when compared to the naïve ones. On 2.2 Proliferative and invasive features of naïve and taxol-resistant TNBC cells, paragraph 1, line 106, the authors affirm a "reduced proliferation capacity" of HCC1806-TaxR cells. Then, lines 112 and 113 state that resistant cells exhibited high proliferative activity compared to untreated control;
Authors: The mentioned lines describe the comparison of the proliferation capacity of HCC1806-TaxR cells in the presence and absence of taxol, not naïve HCC1806 cells. We wanted to hightlight that population growth in the presence of taxol is even more favourable for these resistant cell line. The comparison with naïve HCC1806 cells is described further in lines 118-119.
The text has been corrected to clarify the cell lines being compared (line 115).
Reviewer: Cytotoxicity assay and IC50 determination method are not described in this article;
Authors: Cytotoxicity assay and IC50 determination methods have been added to the “Materials and methods” section of the manuscript as “4.2. Cytotoxicity assay. Cellular Survival MTS-Based Assay”. (lines 441-451).
Reviewer: Line 123: use scientific notation and state the unit (cells/mL, cells/well etc); Lines 531, 533: update scientific notation.
Authors: We made corrections based on the reviewers' recommendations (lines 119, 123,124, 666, 669).
Reviewer: Regarding figures 3 and 4, HCC1806-TaxR cells appears to be bigger (size and nuclei) than HCC1806 naïve, while figure 1D shows cells of similar size. The authors must build panels 3a, 4b and 4e in the same magnifying glass, preferably indicating the structural changes with arrows;
Authors: All images are taken at the same magnification. Previously, we have described the effect of the balance between cytoplasmic β- and γ-actin on cell morphology, where the predominance of γ-actin caused an increase in average cell area (Dugina V, Zwaenepoel I, Gabbiani G, Clément S, Chaponnier C. Beta and gamma-cytoplasmic actins display distinct distribution and functional diversity. J Cell Sci. 2009 Aug 15;122(Pt 16):2980-8. doi: 10.1242/jcs.041970) and in the nuclear area in breast cancer cells (Dugina V, Shagieva G, Novikova M, Lavrushkina S, Sokova O, Kireev I, Kopnin P. Impaired Expression of Cytoplasmic Actins Leads to Chromosomal Instability of MDA-MB-231 Basal-Like Mammary Gland Cancer Cell Line. Molecules. 2021; 26(8):2151. https://doi.org/10.3390/molecules26082151). We observed a predominance of γ-actin in the taxol-resistant cells, that's why HCC1806-TaxR cells appeared larger in size and nuclei than HCC1806 naïve cells. We propose that the increase in nuclear area in cells with γ-actin predominance is caused, at least in part, by enhanced cell spreading.
Reviewer: Authors must update the bibliography. More than 70% of the references are prior to 2016.
Authors: We have added more recent articles to the bibliography of the manuscript.
Reviewer 2 Report
Comments and Suggestions for Authors
In the manuscript entitled „Imbalance between Actin Isoforms Contributes to Tumor Progression in Taxol-Resistant Triple-Negative Breast Cancer Cells” the authors investigate the intricate pathways associated with secondary resistance in triple negative breast cancer (TNBC) cells to taxol. They observed a shift between actin isoforms. The predominance of γ-actin resulted in increased motility and invasiveness. The microtubule system was reorganized as well. It is highly interesting, that taxol resistant breast cancer cells are more effective in forming colonies and in migration. These sentences are very important: „Taxol interacts with β-tubulin isotypes in a unique way. Therefore, a patient's response to the drug may depend in part on the isotype content of the tumour cells… In several cancers, overexpression of βIII-tubulin has been associated with resistance to taxol [28,29,31] Differences in β-tubulin isotypes may be predictive markers for the development of therapeutic strategies with microtubule interacting agents (MIAs).” Which MIAs could be more effective than taxol? Do you have any concrete ideas? If so, please add them to the discussion for this section.
The manuscript is a well-written, thorough work.
The Abstract clearly highlights the main message of the research.
However, two more keywords should be added: microtubule reorganization and pro-metastatic changes.
In contrast to how thoroughly and concisely the whole manuscript is written, the Conclusions are too general. This section should be rewritten. It is important to highlight exactly what new information this research will provide, and exactly how this new knowledge can help cure TNBC.
Author Response
Dear Reviewer,
We sincerely thank you for providing us with the opportunity to improve our manuscript. We have carefully considered all of your suggestions and comments and have made the necessary revisions to improve the presentation of our data. Thank you for your work on our manuscript.
Reviewer: These sentences are very important: „Taxol interacts with β-tubulin isotypes in a unique way. Therefore, a patient's response to the drug may depend in part on the isotype content of the tumour cells… In several cancers, overexpression of βIII-tubulin has been associated with resistance to taxol [28,29,31] Differences in β-tubulin isotypes may be predictive markers for the development of therapeutic strategies with microtubule interacting agents (MIAs).” Which MIAs could be more effective than taxol? Do you have any concrete ideas? If so, please add them to the discussion for this section.
Authors: Epothilones overcome two major types of resistance to taxanes: efflux pump-related resistance and tubulin isotype-related resistance, in particularly connected to the overexpression of βIII-tubulin. However, clinical trials of epothilones have revealed toxicity issues. To address this, we suggest developing therapeutic strategies should evolve towards two- or three-drug combinations (MIAs and individually tailored treatment based on biomarkers). Targeting different cell populations in heterogeneous tumours could resolve toxicity problems and achieve effective tumor control. Deeper classifications of tumours, genomic and proteomic profiles could assist in selecting a specific treatment strategy.
Reviewer: However, two more keywords should be added: microtubule reorganization and pro-metastatic changes.
Authors: We have added the suggested keywords to the appropriate section (lines 39-40).
Reviewer: In contrast to how thoroughly and concisely the whole manuscript is written, the Conclusions are too general. This section should be rewritten. It is important to highlight exactly what new information this research will provide, and exactly how this new knowledge can help cure TNBC.
Authors: We have corrected the 'Conclusions' section in accordance with the reviewer's recommendation (lines 708-721).